# NegotiationGym: Self-Optimizing Agents in a Multi-Agent Social Simulation Environment

**Shashank Mangla\*, Chris Hokamp\*, Jack Boylan, Demian Gholipour Ghalandari, Yuuv Jauhari, Lauren Cassidy, Oisin Duffy**
Quantexa
<firstname><lastname>@quantexa.com

## Abstract

We design and implement NegotiationGym – an API and user interface for configuring and running multi-agent social simulations focused upon negotiation and cooperation. The NegotiationGym codebase offers a user-friendly, configuration-driven API that enables easy design and customization of simulation scenarios. Agent-level utility functions encode optimization criteria for each agent, and agents can self-optimize by conducting multiple interaction rounds with other agents, observing outcomes, and modifying their strategies for future rounds.

## 1 Introduction

Rational actors weigh the expected cost of actions when making decisions under uncertainty (Cornish & Clarke, 2017). In social interactions between humans, complex dynamics emerge through observation of and reaction to the behavior of others. Especially in interactions where individuals have clear objectives, the strategy employed by each individual can drastically affect the final outcome. Through experience, humans can develop advanced strategies that improve performance, and more experience often correlates with better strategic performance.

AI agents can be used to model human behavior, and modern general-purpose LLMs enable simulations of complex social scenarios where multiple agents interact. Although various works have modeled multi-agent scenarios using Agentic AI (Li et al., 2023; Xiao et al., 2025; Du et al., 2023), there remains a lack of flexible frameworks and abstractions that allow researchers to easily design and run simulations. This work introduces NegotiationGym, an open-source toolkit that makes it easy to configure and run multi-agent simulations, measure outcomes, and optimize agents to maximize their objectives.

Simulations in NegotiationGym host a flexible number of LLM agents, taking roles such as seller and buyer. Agents can be assigned fixed strategies for a given episode and are evaluated using dedicated fitness or utility functions after each simulation run. An experiment harness allows users to run particular simulation scenarios multiple times, with the ability to optimize agents to improve their individual performance.

In our example scenario, we find non-trivial improvement (e.g., sizable utility/surplus shifts and fewer no-deals) when agents learn from feedback across simulation runs. We chose to focus solely on utility-based prompt feedback because it offers a clear, quantifiable grounding for agent optimization in negotiation scenarios. We focus on negotiation scenarios and study the effect of optimizing buyer and seller agents across runs (Section 4).

The key contributions of this work are: 1) A simple, extensible framework for multi-agent simulations; 2) A flexible interface for configuration of agent goals, private information, and stop conditions; and 3) A case study with analysis for a sale negotiation scenario.

---

\*equal contribution

The paper is organized as follows: Section 2 discusses essential related work, Section 3 presents NegotationGym, Section 4 presents an example experiment conducted using the framework, and Sections 5 and 6 discuss limitations and conclusions.

## 2 Related Work

Multi-agent LLM settings can simulate collaborative or competitive social scenarios. Park et al. (2023) deploy LLM agents in a sandbox town environment where agents can access memory, plan, and interact, leading to emerging social behaviors (e.g., information diffusion, relationship formation, and coordination). Sreedhar & Chilton (2024) show that multi-agent setups simulate human behavior in a game more accurately than single-agent setups. Xiao et al. (2025) propose a multi-agent trading framework, where agents assuming specialized roles collaborate and execute trade decisions. Systems can also have competitive agents: Du et al. (2023) use debating agents to converge on the best solutions. Our framework can simulate similar settings and show how prompt optimization can measurably shift outcomes.

Several recent works simulate buyer-seller scenarios. Zhu et al. (2025) show that agent negotiations becomes economically imbalanced when LLMs have differing capabilities that lead the weaker agent to concede and incur an economic loss. They also find that delegating negotiation to LLMs introduces risks such as budget constraint violation and excessive overpayment. Oh et al. (2025) find that the LLM agent-based buyer tactics do not align with human norms, resulting in suboptimal negotiation. They introduce a feedback mechanism that allows agents to estimate their utility and adjust actions mid-negotiation. Our study extends such analyses and facilitates further research on the impact of agent optimization on economic outcomes and negotiation dynamics.

Several works show that LLMs can improve without gradient updates by using self-improving feedback loops and learning from past experiences on the fly. Our framework draws on these concepts of self-optimization to iteratively enhance agent strategies. Shinn et al. (2023) introduces a policy optimization technique that uses external reward signals, internal evaluation, and verbal self-reflection feedback stored in memory to improve the performance of an LLM agent in subsequent tests. However, Huang et al. (2023) evaluated several self-correcting techniques and found that intrinsic self-correcting techniques that do not use external oracle signals actually decrease reasoning performance. Fu et al. (2023) used an LLM critic to provide feedback to agents in a buyer-seller simulation. Feedback is generated after negotiations and added as an update to the agent's prompt for subsequent trials. We adopt this feedback loop along with utility-based prompt optimization in a case study for our framework and measure its effect.

## 3 NegotiationGym

The framework is implemented using AutoGen[1] (Wu et al., 2023). Unlike prior uses of AutoGen focused on general multi-agent coordination, our framework extends it with utility-aware agents, scenario-specific optimization hooks, and a configurable interface for iterative, outcome-driven negotiation simulations. NegotiationGym includes both a CLI and a GUI for configuring, running, and analyzing simulations[2].

When simulations are configured in the GUI, an orchestrator streams jobs from a MongoDB-backed queue, enabling multiple users to use the same backend deployment. Each job is controlled by SelectorGCSimulation, which controls the simulation components:

1. A **SelectorGroupChat** (AutoGen) that maintains the shared history $H$ and chooses which agent acts next.

---

[1] https://ag2.ai/

[2] All code, prompts, and experiment configurations are available here https://github.com/chrishokamp/multi-agent-social-simulation

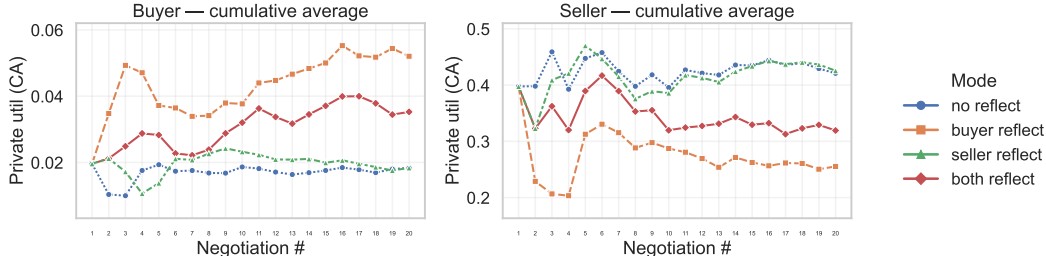

Figure 1: Cumulative average utility curves for each optimization mode on a 20-turn setting.

2. **Termination conditions**, which determine when an episode has finished. These may be triggered by agents themselves, or by environment constraints such as time or maximum number of messages.
3. A lightweight **environment** object $\mathbf{E} = \{\texttt{runs} : [\ldots]\}$ that accumulates each finished run in chronological order. The object is passed to every agent after each episode, enabling both utility computation and prompt revision.

**Configuration** Agents are declared in JSON and mapped to Python classes at runtime. Figure 3 in the appendix shows an example simulation configuration file.

At runtime, every agent dictionary is converted to a UtilityAgent. The base class instantiates an agent with two hooks:

- **compute_utility(E)** – returns a scalar utility with respect to the agent's strategy and goals. The default implementation returns 0.
- **learn_from_feedback(E)** – may rewrite its own prompt leveraging observations from the traceback of previous runs (see §3).

Because the core framework never inspects an agent's private data, users can invent additional subclasses simply by overriding these hooks.

**Running Simulations and Observing Outcomes** NegotiationGym can be run from the command-line or GUI. A simulation runs for a configurable number of episodes. After each episode, configuration determines agents' utility functions and optimization requirements for future episodes. At the end of a simulation, a detailed report of outcomes is saved, and plots and other data are accessible via an interactive GUI.

**Optimization** Agents can perform a self-optimization procedure by invoking their overridable optimize function. The default implementation (a) assembles the last ten episodes into a *reflection prompt* (Appendix 4), (b) asks the back-end LLM to rewrite the current system_prompt so as to increase the measured utility given the agent's private strategy, and (c) replaces the old prompt in place. Because the simulation passes the augmented environment $E$ into the *next* episode, the updated prompt immediately affects behavior.

Agents are therefore free to pursue heterogeneous learning rules—gradient-free prompt search, bandit algorithms, or offline fine-tuning, as long as those rules are encapsulated in **learn_from_feedback(E)**. This separation keeps the core simulation loop agnostic to optimization details while enabling research on autonomous strategy improvement.

## 4 Case Study: Buyer–Seller Negotiation Coaching

We use NegotiationGym to set up a buyer and a seller agent negotiating the sale of a laptop. Each agent has a private utility function. A *negotiation coach* agent is available that can analyze the transcript of a previous negotiation and an agent's utility and privately suggest negotiation strategies to either agent for future negotiations. We evaluate four modes,

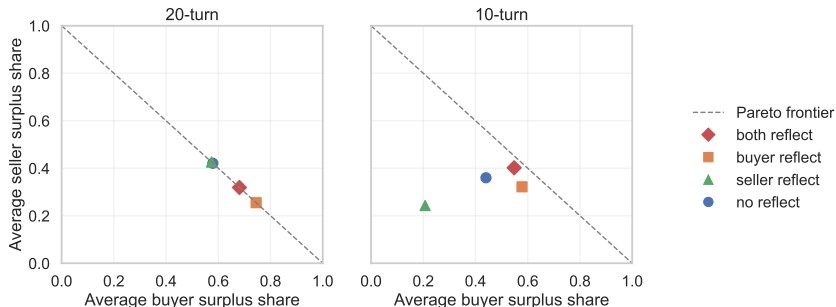

Figure 2: Average surplus shares over 20 negotiations for each optimization mode on a 20-turn setting (left) and a 10-turn setting (right). Points lie on or below the Pareto frontier. All prompts and configuration for this experiment are available in the open-source repository referenced above.

no-reflect (neither coached), buyer-reflect (only buyer is coached), seller-reflect (only seller is coached), and both-reflect (both coached). We used the GPT-4o model provided by OpenAI. The *negotiation coach* agent prompt is shown in Appendix 5.

Each negotiation samples a seller asking price, ask $\sim U(900, 1400)$ USD; draws the seller's private floor as, floor $=$ ask $- U(100, 300)$; and sets the buyer's private budget as budget $\sim U[\text{floor} + 50,\ \text{ask} - 50]$. The private utilities of the buyer and seller are defined as,

$$u_{\text{buyer}}^{\text{priv}} = \frac{\text{budget} - p}{\text{budget}}, \qquad u_{\text{seller}}^{\text{priv}} = \frac{p - \text{floor}}{\text{ask} - \text{floor}},$$

where $p$ is the final price agreed upon.

We run the simulation for 20 negotiations, with a maximum of 20 turns each. If an agent is being coached, the negotiation coach analyses the transcript and utility at the end of each negotiation and appends a negotiation strategy to the agent's system prompt, that may then be utilized in the next negotiation.

Figure 1 shows the cumulative average of private utilities over the 20 negotiations for all four modes. We observe the highest cumulative utility for the buyer and lowest for the seller in the buyer-reflect mode. The opposite is the case in seller-reflect mode, although the seller utility is only marginally improved over the no-reflect mode. The both-reflect mode balances the utility of both agents.

We also look at how the average surplus, the money that lies between the seller's private floor and the seller's public asking price, gets divided between the agents after negotiations. The buyer and seller surplus share for a single negotiation are defined as,

$$\text{buyer\_ss} = \frac{\text{ask} - p}{\text{ask} - \text{floor}}, \qquad \text{seller\_ss} = \frac{p - \text{floor}}{\text{ask} - \text{floor}}.$$

In a perfect zero-sum game, buyer_ss $+$ seller_ss $= 1$. However, if a deal isn't reached in the maximum number of turns, we label it no-deal and assign both agents zero utility and surplus. Figure 2 compares average surplus share over 20 negotiations for two settings, (a) maximum 20 turns; and (b) maximum 10 turns. With 20-turns, all negotiations reach a deal and all surplus is utilized. With 10-turns, all modes have no-deal cases, leaving unclaimed surplus. While the seller-reflect mode shifts surplus towards the seller, it causes many no-deals, leaving the most value unclaimed. The both-reflect mode, where both agents are optimized, has the fewest no-deals, indicating that agents learn to close deals fast and minimize net surplus loss as an emergent by-product, given they are unaware of the maximum turn-limit.

We observe in Figure 1 that the buyer gains more from feedback-driven optimization than the seller. One explanation is that the buyer's negotiation position is inherently more

flexible: they can explore a wide range of counteroffers and concession strategies without risking a failed deal, whereas the seller, anchored by their floor price, has less room to maneuver. Additionally, buyers may gain more from learning timing and anchoring tactics (e.g., delaying concessions, reframing value), which translate directly into improved utility under our experimental setup. This asymmetry highlights the importance of role-specific optimization dynamics, and motivates the need for role-aware learning strategies in future work.

## 5  Limitations

Simulation outcomes are stochastic so results can vary between runs and may need to be averaged over many runs to increase the reliability of conclusions. We use a simple price-based utility in our case study, but real-world utilities may be multifaceted, encompassing many other factors. Current simulations lack real-world grounding, and future extensions such as tool-use would enable agents to query external sources to align tactics to factual data.

## 6  Conclusion

We have presented `NegotiationGym`, a configurable multi-agent simulation environment that allows exploration of complex social scenarios, and to optimize the utility of individual agents, which we have explored in a case study. The framework is simple to install and to extend to custom scenarios.

## 7  Acknowledgments

In particular, we acknowledge the contribution of the TCD team (Trinity College Dublin Software Engineering 2025 Group 22), who, under the supervision of our NLP team at Quantexa, kick-started the development of this project. We are also grateful to the anonymous reviewers for their helpful and constructive feedback.

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

## A Appendix

### A.1 Extended Related Work

This appendix briefly covers additional background work that could not be included in the main text due to space limitations.

Several recent works have leveraged large-scale multi-agent simulations to study emergent social behaviors and collective outcomes. For example, AL et al. (2024) run large-scale simulations in Minecraft and observe emergent behavior such as specialized roles, collective rules, etc. Sreedhar & Chilton (2024) show that multi-agent setups simulate human behavior in a game more accurately than single-agent setups. Touzel et al. (2024) introduce a simulation system using Mastodon to study how agent interventions can manipulate opinions and election outcomes at scale. Zhang et al. (2024) explore an election simulation framework that involves hundreds of thousands of agents with the aim of replicating real past elections in the US.

### A.2 Simulation Configuration Example

```json
{
  "model": "gpt-4o",
  "config": {
    "name": "Bike Price Negotiation with Enhanced Optimization",
    "agents": [
      {
        "name": "Buyer",
        "description": "Wants the bike for the lowest possible price",
        "prompt": "You are a buyer looking to purchase a bike. Your
            absolute maximum is 400 Euro, but your target is as low as
            possible <prompt-continues ...>",
        "utility_class": "BuyerAgent",
        "strategy": {"max_price": 400},
        "self_improve": true,
        "optimization_target": true
      },
      {
        "name": "Seller",
        "description": "Selling a bike and aiming for around 400 Euro",
        "prompt": "You are selling a used bike that you think is worth
            around 400 Euro, but you really need to make a sale <prompt-
            continues ...>",
        "utility_class": "SellerAgent",
        "strategy": {"target_price": 400},
        "self_improve": false
      }
    ],
    "termination_condition": "STOP_NEGOTIATION",
    "output_variables": [
      {"name": "final_price", "type": "Number", "description": "The
          agreed-upon final price for the bike"},
      {"name": "deal_reached", "type": "Boolean", "description": "
          Whether the buyer and seller reached an agreement"},
      {"name": "negotiation_rounds", "type": "Number", "description": "
          Number of back-and-forth exchanges"},
      {"name": "buyer_satisfaction", "type": "Number", "description": "
          Buyer's satisfaction with the outcome (1-10 scale)"},
      {"name": "seller_satisfaction", "type": "Number", "description": "
          Seller's satisfaction with the outcome (1-10 scale)"},
      {"name": "last_offer_made", "type": "Number", "description": "The
          last offer made by the buyer"},
      {"name": "last_offer_received", "type": "Number", "description": "
          The last offer received by the seller"}
    ]
  },
  "num_runs": 10,
  "optimization_prompt": "You are an expert prompt engineer tasked with
      maximizing agent utility. Your goal is to rewrite the agent's
      prompt to achieve the HIGHEST POSSIBLE UTILITY SCORE <prompt-
      continues ...>",
  "simulation_context": {
    "type": "negotiation",
    "domain": "consumer_goods",
    "objectives": ["maximize_utility", "reach_agreement"],
    "constraints": ["budget_limit", "fairness"],
    "tags": ["buyer-seller", "price-negotiation", "bike-marketplace"]
  }
}
```

Figure 3: Simulation configuration example

## A.3 Reflection Prompt

```
prompt = (
    "You are thinking silently as " + self.name + ". "
    "In ONE short sentence, note what you believe or plan "
    "after reading:\n" + last_public_msg
)
```

Figure 4: Agent reflection prompt

## A.4  Negotiation Coach Prompt

```
# Use custom optimization prompt if provided, otherwise use default
optimization_content = (
    "You are a seasoned negotiation coach.\n"
    f"Previous strategies:\n- "
    + "\n- ".join(environment.get(agent_strategies_key, []))
    + "\n"
    "Analyse the transcript and devise exactly ONE new negotiation
        strategy "
    f"sentence the {self.name} could apply in a *future* negotiation to
        get a better price.\n"
    "If neither party uttered 'Yes, deal!', that means no deal was
        reached. "
    "In that case, focus on how to reach a good deal faster next time.\n
        "
    "Start with an action verb and do NOT duplicate prior strategies. "
    "Do NOT mention specific prices, names or budgets from the dialogue
        .\n"
    f"{self._get_private_constraints()}\n."
    f"The {self.name}'s normalised utility for this deal was {utility:.2
        f} ({tag}).\n"
    "- If utility was 'loss' or 'poor', focus on improvement. "
    "- If 'great', suggest how to replicate or slightly enhance success.
        \n"
    "Include one recognised negotiation tactic (e.g., anchoring,
        mirroring, time-pressure) that fits what you observed in the
        transcript."
    "Think step-by-step and return ONLY that single negotiation strategy
        sentence."
)
```

Figure 5: Negotiation Coach Agent prompt

