# OpenReview forum: "NegotiationGym: Self-Optimizing Agents in a Multi-Agent Social Simulation Environment"
_colmweb.org/COLM/2025/Workshop/Social_Sim — Social Sim'25_

### Official Review · Reviewer_soYY · 2025-07-11

**Rating:** 7
**Overall Assessment:** 4
**Confidence:** 3

**Review:**

The paper is overall clearly written and it is straight to the point: their framework is simple but flexible enough to perform diverse experiments, and the authors present an example of buyer-seller negotiations. Another pro is that the code is already online, and the  provided example is clear enough. Attendees of the workshop will benefit from the framework described in this paper.

Nonetheless, I have list of things to point out to improve the paper:
- Lines 65-66: Provide a brief motivation about why you decided to choose utility-based prompt feedback only, as opposed to other types of feedback.
- Line 68: So it seems the paper's framework uses AutoGen as its backbone. Add a brief description about how the author's framework differentiates from the AutoGen paper's framework and others that have used it in the past.
- Line 83: Oddly enough, there is a reference to "(Alg. 1, line 3)", however, I don't see such Algorithm in the paper! Please, add the algorithm.
- In the buyer-seller case: Why do you think the buyer benefits more from feedback than the seller? At least that is why I see in Figure 1. It will be really helpful, even to address this question, to include the prompts used for the experiment in the appendix.
- What is the "reflection prompt" from line 98? Please, add to Appendix. Also add the prompt given to the "negotiation coach" agent.
- The footnote in page 2 mentions the link has access to "video", but I couldn't find any video. What is the "video" about?

Other lesser things:
- Line 29: specify what is "non-trivial" about the utility improvement in your example of buyer-seller.
- Line 41: what kind of "emerging social behaviors" are you referring to?
- Line114: "... private budget as budget ... " (eliminate the comma).

**Comments Suggestions And Typos:**

See "Review" section.

**Paper Summary:**

The paper is within the scope of the workshop. The paper presents an environment for multiple agents to interact according to (i) some internal utility they seek to optimize and (ii) the possibility of incorporating feedback into their prompt to modify their behavior. It uses AutoGen as part of its backend and allows the possibility of developers to modify the integration of signals that can modify agent behavior besides (i) and (ii).

**Relevance:**

5

**Summary Of Strengths:**

See "Review" section.

**Summary Of Weaknesses:**

See "Review" section.

---

### Meta-Review · Area_Chair_1nQK · 2025-07-21

**Recommendation:** Accept

**Metareview:**

--